# Learning to Encode Text as Human-Readable Summaries using Generative Adversarial Networks

## Abstract

Auto-encoders compress input data into a latent-space representation and reconstruct the original data from the representation. This latent representation is not easily interpreted by humans. In this paper, we propose training an auto-encoder that encodes input text into human-readable sentences. The auto-encoder is composed of a generator and a reconstructor. The generator encodes the input text into a shorter word sequence, and the reconstructor recovers the generator input from the generator output. To make the generator output human-readable, a discriminator restricts the output of the generator to resemble human-written sentences. By taking the generator output as the summary of the input text, abstractive summarization is achieved without document-summary pairs as training data. Promising results are shown on both English and Chinese corpora.

## 1 Introduction

When it comes to learning data representations, a popular approach involves the auto-encoder architecture, which compresses the data into a latent representation without supervision. In this paper we focus on learning text representations. Because text is a sequence of words, to encode a sequence, a sequence-to-sequence (seq2seq) auto-encoder (Li et al., 2015; Kiros et al., 2015) is usually used, in which a RNN is used to encode the input sequence into a fixed-length representation, after which another RNN is used to decode the original input sequence given this representation.

Although the latent representation learned by the auto-encoder can be used in downstream applications, they are usually not human-readable. In this work, we use comprehensible natural language as a latent representation of the input source text in an auto-encoder model. This human-readable latent representation is shorter than the source text; in order to reconstruct the source text, it must reflect the core idea of the source text. Intuitively, the latent representation can be considered a summary of the text.

The idea that using human comprehensible representation as a latent representation has been explored on text summarization (Miao & Blunsom, 2016), but only in a semi-supervised scenario. Previous work uses a prior distribution from a pre-trained language model to constrain the generated sequence to natural language. However, to teach the compressor network to generate text summaries, the model is trained using labeled data. In contrast, in this work we need no labeled data to learn the representations.

The proposed model is inspired from cycle consistency (Zhu et al., 2017; He et al., 2016). As shown in Fig. 1, the proposed model is composed of three components: a generator, a discriminator, and a reconstructor. Together, the generator and reconstructor form a text auto-encoder. The generator acts as an encoder in generating the latent representation from the input text. Instead of using a vector as latent representation, however, the generator generates a word sequence much shorter than the input text. From the shorter text, the reconstructor reconstructs the original input of the generator. By minimizing the reconstruction errors, the generator learns to generate short text segments that contain the main information in the original input. We use the seq2seq model in modeling the generator and reconstructor because both have input and output sequences with different lengths.

However, it is very possible that the generator's output word sequence can be processed by the reconstructor but is not readable by humans. Here, instead of regularizing the generator output

with a pre-trained language model (Miao & Blunsom, 2016), we borrow from adversarial auto-encoders (Makhzani et al., 2015) and introduce a third component – the discriminator – to regularize the generator's output word sequence.

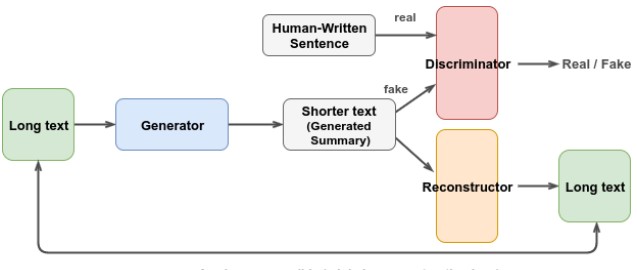

Figure 1: Proposed model. Given long text, the generator produces a shorter text as a summary. The generator is learned by minimizing the reconstruction loss together with the reconstructor and making discriminator regard its output as human-written text.

The discriminator and the generator form a generative adversarial network (GAN) (Goodfellow et al., 2014). GANs are generative models composed of a generator and a discriminator. The discriminator discriminates between the generator output and real data, and the generator produces output as similar as possible to real data to confuse the discriminator. Here, we only have to feed human-written sentences to the discriminator as real data. With the GAN framework, the discriminator teaches the generator how to create human-like summary sentences as a latent representation; however, this only guarantees that the generator produces grammatically correct sentences – not necessarily sentences that represent the input text. It is the reconstructor that teaches the generator how to produce a sentence that captures the core idea of the source text.

However, generating discrete distributions with GAN is challenging, since it is difficult to evaluate the distance between the continuous distribution from the generator and the discrete distribution of the real sample. In addition, if we feed sampled words from the generator output distribution to the discriminator, the process of word selection is non-differentiable, which yields a discriminator gradient that precludes back-propagation to the generator. With GAN, there are two ways to generate language: (1) by training with a policy gradient, which regards words as actions, or (2) by directly feeding the generator's output layer to the discriminator, which yields a gradient suited to back-propagation to the generator. In this work, we propose new kind of method on training with policy gradient in which the discriminator evaluates the output of generator every time steps. On language generation with GAN, we conduct experiments using both (1) and (2) methods and evaluate their results.

We evaluate the results on an abstractive text summarization task in which the machine generates a text summary in its own words. The model is learned from a set of unpaired documents and summaries[1]. We use the sentences in the summaries as real data for discriminator[2]. As the summaries can come from another set of documents not related to the training documents, training is unsupervised. We use the output word sequence of the generator as the summaries of the input text. The results show that the generator generates summaries with reasonable quality on both English and Chinese corpora.

## 2 RELATED WORK

### GAN FOR LANGUAGE GENERATION

The major challenge in applying GAN to sentence generation is the discrete nature of natural language. To generate a word sequence, the generator usually has non-differential parts such as *argmax*

---

[1]Here the titles of the documents are considered as the summaries. This is a typical setup in the study of summarization.

[2]Instead of using general sentences as real data for discriminator, we here choose sentences from summaries because they have their own unique distribution. For example, the structure of a summary sentence is concise; they usually contain specific words such as country or person names, or key verbs from the source text. Because the following application is document summarization, using document summaries as real data is helpful. It is possible to use text from different styles as real data for the discriminator to produce output from the generator in different styles.

or other sample functions which cause the original GAN to fail. Therefore, new kinds of GANs have been proposed for sentence generation.

SeqGAN (Yu et al., 2017) tackles the sequence generation problem with reinforcement learning. Here, we refer to this approach as adversarial REINFORCE, in which the generator is regarded as an agent, the generated sequence of words is viewed as a sequence of actions, and the current state is defined as the generated sequence to date and the prior input. However, the discriminator only measures the quality of whole sentences, and thus the rewards are extremely sparse and the rewards assigned to all actions in sequence are all the same. To tackle this problem, they propose MC search to evaluate approximate rewards at each time step, but this method suffers from high time complexity. Following this idea, (Li et al., 2017) proposes another approach to evaluate the expected reward at each time step. They break both the generated and real sequences into partial sequences, and the discriminator discriminates between the generated and real partial sequences. Inspired by this idea, we propose the self-critical adversarial REINFORCE algorithm as another way to evaluate the expected reward at each time step.

In (Gulrajani et al., 2017), instead of feeding a discrete word sequence, the authors directly feed the generator output layer to the discriminator. This method works because they use the earth mover's distance on GAN as proposed in (Arjovsky et al., 2017), which is able to evaluate the distance between a discrete and a continuous distribution. In order to satisfy the requirement of the earth mover's distance, they use a gradient penalty trick to confine the complexity of discriminator function. Their method achieves an amazing result: it is the first work on GAN training that performs language generation without pre-training. In our work, we also conduct experiments on this method with discriminator settings almost the same as the original paper.

### ABSTRACTIVE TEXT SUMMARIZATION

Recent model architectures for abstractive text summarization basically use the sequence-to-sequence (Sutskever et al., 2014) framework in combination with various novel mechanisms. One popular mechanism is attention (Bahdanau et al., 2015), which has been shown helpful for summarization (Nallapati et al., 2016; Rush et al., 2015). It is also possible to directly optimize evaluation metrics such as ROUGE (Lin, 2004) with reinforcement learning (Ranzato et al., 2016; Paulus et al., 2017; Bahdanau et al., 2016). The hybrid pointer-generator network (See et al., 2017) selects words from the original text with a pointer (Vinyals et al., 2015) or from the whole vocabulary with a trained weight. In order to eliminate repetition, a coverage vector (Tu et al., 2016) can be used to keep track of attended words and coverage loss (See et al., 2017) can be used to encourage model focus on diverse words. While most papers focus on supervised learning with novel mechanisms, we explore unsupervised training models.

## 3 PROPOSED METHOD

The overview of the proposed model is shown in Fig. 2. The model is composed of three components: generator $G$, discriminator $D$, and reconstructor $R$. Both $G$ and $R$ are seq2seq hybrid pointer-generator networks (See et al., 2017) which can decide to copy words from encoder input text via pointing or generate from vocabulary.They both take a word sequence as input and output a sequence of word distributions. Discriminator $D$, on the other hand, takes a sequence as input and outputs a scalar. The model is learned from a set of documents $x$ and human-written sentences $y^{real}$. Although in real implementation, $y^{real}$ are the sentences in summaries, we note that the documents and summaries are unpaired.

To train the model, a training document $x = \{x_1, x_2, ..., x_t, ..., x_T\}$, where $x_t$ represents a word, is fed to $G$, which outputs a sequence of word distributions $G(x) = \{y_1, y_2, ..., y_n, ..., y_N\}$, where $y_n$ is a distribution over all words in the lexicon. Then we sample a word $y_n^s$ from each distribution $y_n$, and a word sequence $y^s = \{y_1^s, y_2^s, ..., y_N^s\}$ is obtained according to $G(x)$. We feed the sampled word sequence $y^s$ to reconstructor $R$, which outputs another sequence of word distributions $\hat{x}$. The reconstructor $R$ reconstructs the original text $x$ from $y^s$. That is, we seek an output of reconstructor $\hat{x}$ that is as close to the original text $x$ as possible; hence the loss for training the reconstructor $R$, $R_{loss}$, is defined as

$$R_{loss} = \sum_{k=1}^{K} l_s(x, \hat{x}), \tag{1}$$

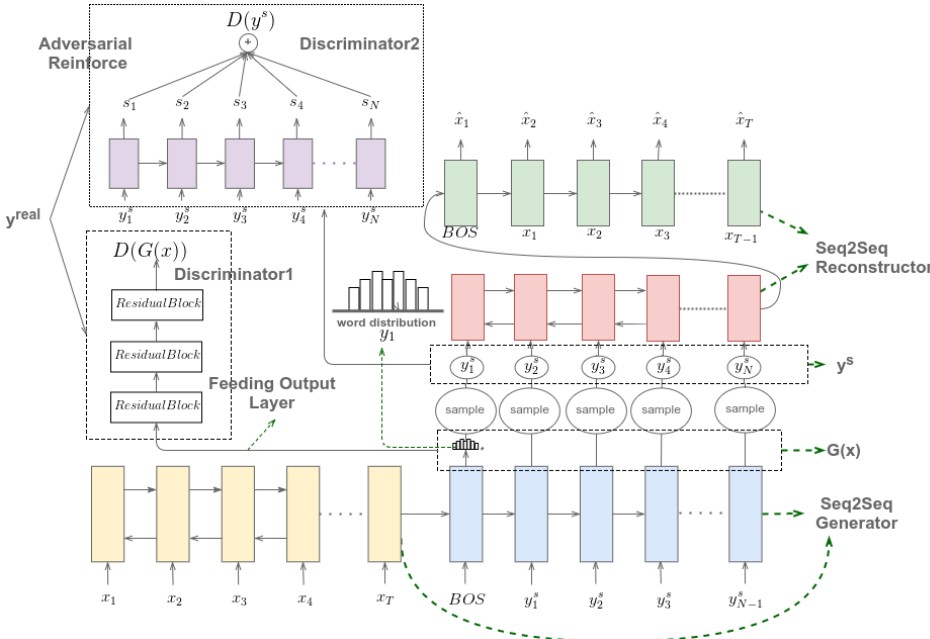

Figure 2: Architecture of proposed model. The generator network and reconstructor network are a seq2seq hybrid pointer-generator network, but for simplicity, we omit the pointer and the attention parts.

where the reconstruction loss $l_s(x, \hat{x})$ is the cross-entropy loss computed between the reconstructor output sequence $\hat{x}$ and the source text $x$, or the negative conditional log-likelihood of source text $x$ given word sequence $y^s$ sampled from $G(x)$. The reconstructor output sequence $\hat{x}$ is teacher-forced by source text $x$. The subscript $_s$ in $l_s(x, \hat{x})$ indicates that $\hat{x}$ is reconstructed from $y^s$. $K$ is the number of training examples (documents), and (1) is the summation of the cross-entropy loss over all the training documents $x$.

In the proposed model, the generator $G$ and reconstructor $R$ form an auto-encoder. However, the reconstructor $R$ does not directly take the generator output distribution $G(x)$ as input [3]. Instead, the reconstructor takes a sampled discrete sequence $y^s$ as input. Due to the non-differentiable property of discrete sequences, we apply the REINFORCE algorithm, which is described in Section 4.

In addition to reconstruction, we need the discriminator $D$ to discriminate between the real sequence $y^{real}$ and the generated sequence $y^s$ to regularize the generated sequence satisfying the summary distribution. $D$ learns to give $y^{real}$ higher scores while giving $y^s$ lower scores. The loss for training the discriminator $D$ is denoted as $D_{loss}$; this is further described in Section 5.

$G$ learns to minimize the reconstruction error $R_{loss}$, while maximizing the loss of the discriminator $D$ by generating a summary sequence $y^s$ that cannot be differentiated by $D$ from the real thing. The loss when training the generator $G$, $G_{loss}$, is

$$G_{loss} = \alpha R_{loss} - D'_{loss} \tag{2}$$

where $D'_{loss}$ is highly related to $D_{loss}$ – but not necessary the same – and $\alpha$ is a hyper-parameter. After obtaining the optimal generator by minimizing (2), we use it to generate summaries.

Generator $G$ and discriminator $D$ together form a GAN. We use two different adversarial training methods to train $D$ and $G$; as shown in Fig. 2, these two methods have their own discriminators 1 and 2. Discriminator 1 takes the generator output layer $G(x)$ as input, whereas discriminator 2 takes the sampled discrete word sequence $y^s$ as input. The two methods are described respectively in Sections 5.1 and 5.2.

---

[3]We found that if the reconstructor $R$ directly takes $G(x)$ as input, the generator $G$ learns to put the information about the input text in the distribution of $G(x)$, making it difficult to sample meaningful sentences from $G(x)$.

## 4 MINIMIZING RECONSTRUCTION ERROR

Because discrete sequences are non-differentiable, we use the REINFORCE algorithm. The generator is seen as an agent whose reward given the source text $x$ is $-l_s(x, \hat{x})$. Maximizing the reward is equivalent to minimizing the reconstruction loss $R_{loss}$ in (1). However, the reconstruction loss varies widely from sample to sample, and thus the rewards to the generator are not stable either. Hence we add a baseline to reduce their difference. We apply self-critical sequence training (Rennie et al., 2017); the modified reward $r^R(x, \hat{x})$ from reconstructor $R$ with the baseline for the generator is

$$r^R(x, \hat{x}) = -l_s(x, \hat{x}) - (-l_a(x, \hat{x}) - b) \tag{3}$$

where $-l_a(x, \hat{x}) - b$ is the baseline. $l_a(x, \hat{x})$ is also the same cross-entropy reconstruction loss as $l_s(x, \hat{x})$, except that $\hat{x}$ is obtained from $y^a$ instead of $y^s$. $y^a$ is a word sequence $\{y_1^a, y_2^a, ..., y_n^a, ..., y_N^a\}$, where $y_n^a$ is selected using the *argmax* function from the output distribution of generator $y_n$. As in the early training stage, the sequence $y^s$ barely yields higher reward than sequence $y^a$, to encourage exploration we introduce the second baseline score $b$, which gradually decreases to zero. Then, the generator is updated using the REINFORCE algorithm with reward $r^R(x, \hat{x})$ to minimize $R_{loss}$.

## 5 GAN TRAINING

With adversarial training, the generator learns to produce sentences as similar to the human-written sentences as possible. Here, we conduct experiments on two kinds of methods of language generation with GAN. In Section 5.1 we directly feed the generator output probability distributions to the discriminator and use a Wasserstein GAN (WGAN) with a gradient penalty. In Section 5.2, we explore adversarial REINFORCE, which feeds sampled discrete word sequences to the discriminator and evaluates the quality of the sequence from the discriminator for use as a reward signal to the generator.

### 5.1 DISCRIMINATOR 1: WASSERSTEIN GAN

In the lower left of Fig. 2, the discriminator model is shown as **discriminator1**. The discriminator loss $D_{loss}$ is

$$D_{loss} = \frac{1}{K} \sum_{k=1}^{K} D(y^{s(k)}) - \frac{1}{K} \sum_{k=1}^{K} D(y^{real(k)}) + \beta (\Delta_{y^{i(k)}} D(y^{i(k)}) - 1)^2, \tag{4}$$

where $K$ denotes the number of training examples in a batch, and $k$ denotes the $k$-th example. The last term in (4) is the gradient penalty (Gulrajani et al., 2017). We interpolate the generator output layer $G(x)$ and the real sample $y^{real}$, and apply the gradient penalty to the interpolated sequence $y^i$. $\beta$ determines the gradient penalty scale. In Equation (2), for WGAN, $D'_{loss}$ is the score of the generated example:

$$D'_{loss} = \frac{1}{K} \sum_{k=1}^{K} D(G(x^{(k)})).$$

### 5.2 SELF-CRITIC ADVERSARIAL REINFORCE

In this section, we describe in detail the proposed adversarial REINFORCE method. The core idea is we use the LSTM discriminator to evaluate the current quality of the generated sequence $\{y_1^s, y_2^s, ..., y_i^s\}$ at each time step $i$. Hence, the generator knows that compared to the last time step, as the generated sentence either improves or worsens, it can easily find the problematic generation step in a long sequence, and thus fix the problem easily.

#### 5.2.1 DISCRIMINATOR 2

As shown in Fig. 2, the **discriminator2** is a one-way LSTM network which takes a discrete word sequence as input. At time step $i$, given input word $y_i^s$ it predicts the current score $s_i$ based on the sequence $\{y_1, y_2, ..., y_i\}$. The score is viewed as the quality of the current sequence.

In order to compute the discriminator loss $D_{loss}$ , we sum the scores $\{s_1, s_2, ..., s_N\}$ of the whole sequence $y^s$ to yield

$$D(y^s) = \frac{1}{N} \sum_{n=1}^{N} s_n.$$

$$D_{loss} = \frac{1}{K} \sum_{k=1}^{K} D(y^{s(k)}) - \frac{1}{K} \sum_{k=1}^{K} D(y^{real(k)}),$$

where $K$ and $N$ denote the number of training examples and the generated sequence length respectively. With the loss mentioned above, the discriminator attempts to quickly determine whether the current sequence is real or fake. The earlier the timestep discriminator determines whether the current sequence is real or fake, the lower its loss. An example is shown in Fig. 3.

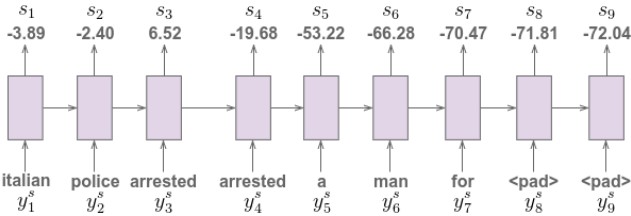

Figure 3: When the second *arrested* appears, the discriminator determines that this example came from the generator. Hence, after this time-step, it outputs low scores.

### 5.2.2 SELF-CRITICAL GENERATOR

Since we feed a discrete sequence $y^s$ to the discriminator, the gradient from the discriminator cannot directly back-propagate to the generator. Here, we use the policy gradient method. At timestep $i$, we use the $i - 1$ timestep score $s_{i-1}$ from the discriminator as its self-critical baseline. The reward $r_i^D$ evaluates whether the quality of sequence in timestep $i$ is better or worse than that in timestep $i - 1$. The generator reward $r_i^D$ from $D$ is

$$r_i^D = \begin{cases} s_i & \text{if i = 1} \\ s_i - s_{i-1} & \text{otherwise.} \end{cases}$$

However, some sentences may be judged as bad sentences at the previous timestep, but at later timesteps judged as good sentences, and vice versa. Hence we use the discounted expected reward $d$ with discount factor $\gamma$ to calculate the discounted reward $d_i$ at time step $i$ as

$$d_i = \sum_{j=i}^{N} \gamma^{j-i} r_j^D.$$

The adversarial REINFORCE score related to discriminator $D'_{loss}$ in (2) is

$$D'_{loss} = E_{y_i^s \sim p_G(y_i^s | y_1^s, ..., y_{i-1}^s, x)}[d_i].$$

We use the likelihood ratio trick to approximate the gradient.

## 6 IMPLEMENTATION

**Network Architecture.** The model architecture of generator and reconstructor is almost same except the length of input and output sequence. We adapt model architecture for our generator and reconstructor from See et al. (2017) who used hybrid-pointer network with coverage vector for text summarization. The hybrid-pointer networks of generator and reconstructor are all composed of two one-layer LSTMs as its encoder and decoder, respectively, with a hidden layer size of 600. Since we use two kinds of methods on adversarial training, there are two discriminators with different model architecture. In the Section 5.1, the discriminator is composed of four residual blocks with 512 hidden dimensions. While in Section 5.2, we use only one hidden-layer one-way LSTM with a hidden size of 512 as our discriminator.

**Details of Training.** We set the weight $\alpha$ in (2) controlling $R_{loss}$ to 10 if not specified. We find that the if the value of $\alpha$ is too large, generator will start to generate output unlike human-written

sentences. On the other hand, if the value of $\alpha$ is too small, the sentences generated by generator will sometimes become unrelated to input text of generator. For all the experiments, the baseline $b$ in (3) gradually decreases from 0.25 to zero within 10000 updates on generator.

In Section 5.1, we set the weight $\beta$ of the gradient penalty to 10, and used RMSPropOptimizer with a learning rate of 0.00001 and 0.001 on the generator and discriminator, respectively. In Section 5.2, we clip the value of the weights of discriminator to $\pm0.15$, and used RMSPropOptimizer with a learning rate of 0.0001 and 0.001 on the generator and discriminator, respectively. It's also feasible to apply gradient penalty trick in this method to satisfy requirement of Wasserstein distance. However, in this method, the performance of gradient penalty trick and weights clipping trick is close.

## 7 EXPERIMENT

We evaluate our model on the Chinese Gigaword and English Gigaword datasets. Before jointly training the whole model, we pre-trained the three major components – generator, discriminator, and reconstructor – separately. First, we pre-trained the generator in an unsupervised manner so that the generator would be able to somewhat grasp the semantic meaning of the source text. The details of the pre-training are in Appendix A. We pre-trained the discriminator and reconstructor respectively with the pre-trained generator's output to ensure that these two critic networks provide good feedback to the generator. During testing, when using the generator to generate summaries, we simply selected the words in a greedy fashion without beam-search, and we eliminated repetition.

### 7.1 CHINESE GIGAWORD

| Methods | | ROUGE-1 | ROUGE-2 | ROUGE-L |
|---|---|---|---|---|
| (A) Training with paired data (supervised) | | 48.664 | 33.907 | 45.685 |
| (B) Trivial baselines: lead - 15 | | 30.077 | 18.237 | 27.736 |
| (C) Unsupervised | (C-1) Pretrained generator | 28.122 | 16.656 | 26.227 |
| | (C-2) **WGAN** | 37.803 | 24.460 | 35.116 |
| | (C-3) **Adversarial REINFORCE** | 40.053 | 26.126 | 37.118 |

Table 1: Results on Chinese Gigaword. In row (B), we select the article's first fifteen words as its summary. Part (C) are the results obtained without paired data.

The Chinese Gigaword corpus is composed of 2.2M paired data of headlines and news. We preprocessed the raw data as following. First, we selected the 4000 most frequent Chinese characters as our vocabulary. We filtered out headline-news pairs with excessively long or short news segments, or that contained too many out-of-vocabulary Chinese characters, yielding 1.1M headline-news pairs from which we randomly selected 5K headline-news pairs as our testing set, 5K headline-news pairs as our validation set, and the remaining pairs as our training set. During training and testing, the generator took only the first 80 Chinese characters of the source text as input.

The results are shown in Table 1. Row (A) lists the results using 1.1 million document-summary pairs to directly train the generator without the reconstructor and discriminator: this is the upper bound of the proposed approach. In row (B), we simply took the first fifteen words in a document as its summary. The number of words was chosen to optimize the evaluation metrics. Part (C) are the results obtained in the unsupervised scenario without paired data. We show the results of the pre-trained generator in row (C-1); rows (C-2) and (C-3) are the results for the two GAN training methods respectively. We find that despite the performance gap between the unsupervised and supervised methods (rows (C-2), (C-3) v.s. (A)), the proposed method yielded much better performance than the trivial baselines (rows (C-2), (C-3) v.s. (B)).

### 7.2 ENGLISH GIGAWORD

On English Gigaword, we set our vocabulary size to 15k, and used the dataset preprocessed by (Rush et al., 2015) for training and testing. We used 3.8M unpaired training data for our training, and we used the whole 200k filtered data in validation set for testing.

The results on English Gigaword are shown in Table 2. In row (B-1), we simply took the first eight words in a document as its summary. Row (B-2) is another trivial baseline. With unpaired documents and summaries, we matched documents to the most relevant summaries with unsupervised method. Each document and each summary were represented as tf-idf (term frequency & inverse document

| Methods | | ROUGE-1 | ROUGE-2 | ROUGE-L |
|---|---|---|---|---|
| (A) Training with paired data (supervised) | | 37.469 | 16.272 | 35.175 |
| (B) Trivial baselines | (B-1) Lead-8 | 27.663 | 10.246 | 25.852 |
| | (B-2) Unsupervised matching | 29.900 | 10.442 | 27.379 |
| (C) Unsupervised | (C-1) Pre-trained generator | 21.269 | 5.608 | 18.896 |
| | (C-2) **WGAN** | 33.043 | 12.222 | 30.121 |
| | (C-3) **Adversarial REINFORCE** | 32.826 | 9.332 | 28.727 |
| (D) Transfer learning (Pre-train) | (D-1) Pre-trained generator | 22.540 | 6.652 | 20.879 |
| | (D-2) **WGAN** | 32.405 | 12.313 | 29.689 |
| | (D-3) **Adversarial REINFORCE** | 31.487 | 10.495 | 28.248 |
| (E) Transfer learning (Pre-train+Discriminator) | (E-1) **WGAN** | 29.912 | 10.695 | 27.324 |
| | (E-2) **Adversarial REINFORCE** | 27.755 | 9.280 | 24.860 |

Table 2: Results on English Gigaword: In row (B-1), we select the article's first eight words as its summary. In row (B-2), we match the documents to their most relevant summaries with unsupervised method. Part (C) are the results obtained without paired data. In part (D), we pre-trained the generator on CNN/Diary. In part (E), we not only pre-trained on CNN/Diary but also used the summaries from CNN/Diary as real data for the discriminator.

frequency) vectors. The summary whose vector maximized cosine similarity of a document vector was retrieved as summary of the document. With paired data from unsupervised matching, given documents as generator input, the generator was trained to predict retrieved summaries.

The results for the pre-trained generator is shown in row (C-1). Compared with the trivial baselines (part (B)), the proposed approach (rows (C-2) and (C-3)) showed good improvement in terms of ROUGE-1. As shown in Fig. 4, the unsupervised method selects key words in the source text and generates the text summary. However, as shown in Table 2, although both unsupervised methods yield ROUGE-1 scores close to that of supervised training, they achieve lower scores on ROUGE-2, especially when training GAN with reinforcement learning. This is because they are extracting the key words from the source text, despite sometimes failing to arrange these words in the correct order. As shown in part (C-3) of Fig. 5, the words in the sentence generated in an unsupervised manner are not arranged correctly: the Italian prime minister, Berlusconi, should not be visiting himself.

## 7.3 TRANSFER LEARNING

In this subsection, we study transfer learning. We used the CNN/Daily Mail dataset (Hermann et al., 2015; Nallapati et al., 2016) preprocessed by the script provided by (See et al., 2017) as our source domain $S$, and English Gigaword as our target domain $T$. The data distributions among the two datasets are quite different. In English Gigaword, the articles consist of 32 words on average and the summaries consist of one sentence with 8 words on average, whereas the CNN/Daily Mail dataset is composed of 790-word articles and multi-sentence summaries. We took only the first 30 to 45 words in the original source domain articles as our new source articles $S_a$. There are 50K source articles in $S_a$. The 50K source summaries of the source articles are $S_t$. In contrast to English Gigaword, the summaries in CNN/Daily Mail dataset are composed of more than one sentence. We split the summary of each article into several sentences, and obtained 240K sentences $S_r$ in this way.

Transfer learning was applied in two directions. In the first direction, we pre-trained our generator on the data from the source dataset: we pre-trained the generator with $S_a$ as input, and the generator predicted $S_t$. We pre-trained the generator in this manner in all of the transfer learning experiments. Then, after pre-training, the generator was further learned jointly with the reconstructor and discriminator on the data from the target domain. The results are shown in part (D) of Table 2. We found that pre-training on the source data set did not degrade performance, and even improved performance in some cases (parts (D) v.s. (C)).

In the second direction, we used the summary from the source domain as our real data. The experiments conducted up to this point required unpaired summary and text data in the target domain. In this experiment, the task was more challenging in that we used summaries $S_r$ from the source domain $S$ as the real data for the discriminator; the generator took the target domain text as input. However, the summaries in each summarization task dataset had a different distribution in terms of the writing style, or in terms of the preferred summary words. To prevent overfitting to $S_r$, we set

the weight $\alpha$ to 50 which was larger than other experiments. With small weight of $\alpha$, as training progressed, the generator summary diverged more and more from the article, and the ROUGE scores became lower and lower.

The results without using summaries in the target domain are shown in part (E). We find that using sentences $S_r$ from another dataset yields lower ROUGE scores on the target testing set (parts (E) v.s. (D)) due to the mismatch between the summaries of the source and target domains. However, the discriminator still roughly regularizes the language model of generated word sequence. After training, the model still greatly enhanced the ROUGE score of the pre-trained model (rows (E-1), (E-2) v.s. (D-1)).Although the results in part (E) are comparable with the trivial baselines in part (B), in inspecting the real examples, we found that the results in part (E) were in fact better; this is not reflected in the ROUGE scores. In Fig. 4, the results in part (E) are better than the leading 8 words and the pre-trained generator results. To support this idea, we provide more examples in the Appendix C.

| Source Text: | |
|---|---|
| three stores and markets in beijing 's fengtai district have been forced to shut down and yesterday each was fined ###,### yuan -lrb- ##,### us dollars -rrb- for violating laws and regulations on fire prevention and control . | |
| **Ground Truth:** stores markets punished for lack of fire controls | **(A)Supervised Result:** beijing 's district stores shut down |
| **(C-1)Pretrained Generator:** and yesterday prevention have been forced to shut down | **(D-1)Pretrained Generator:** three stores and markets fined ###,### yuan dollars |
| **(C-2)WGAN:** three stores markets in beijing 's district have forced to shut down yesterday | **(C-3)Adversarial REINFORCE:** three stores in beijing forced to shut down for violating regulations |
| **(E-1)WGAN :** three stores and markets was forced to shut down | **(E-2)Adversarial REINFORCE:** three stores and markets was forced to shut down and yesterday each was fined |

Figure 4: Real example from our model in English Gigaword. The proposed method generates summaries that capture the core idea of the article.

| Source Text: | |
|---|---|
| italian prime minister silvio berlusconi arrived monday in the libyan capital tripoli , one of only a few high-ranking european officials to visit the country in the last \#\# years . | |
| **Ground Truth:** italy 's prime minister arrives in libya ; highest-ranking european to visit in \#\# years | **(A)Supervised Result:** berlusconi arrives in libya |
| **(C-1)Pretrained Generator:** the libyan capital in tripoli to visit the country | **(D-1)Pretrained Generator:** , has been trying to visit the country from tripoli |
| **(C-2)WGAN:** italian visit in libyan capital tripoli | **(C-3)Adversarial REINFORCE:** italian prime minister to visit berlusconi in tripoli years |
| **(E-1)WGAN :** italian prime minister silvio berlusconi arrived in libyan capital tripoli | **(E-2)Adversarial REINFORCE:** arrived in the libyan capital tripoli |

Figure 5: In part (C-3), some words in the summary sentences are arranged in incorrect order.

## 7.4 SEMI-SUPERVISED LEARNING

In semi-supervised training, generator was pre-trained with few available labeled data, and during unsupervised training, we conducted teacher-forcing with labeled data on generator every several unsupervised updates. In teacher forcing, given source text as input, the generator was teacher-forced to predict the human-written summary of source text. Teacher-forcing can be regarded as regularization of unsupervised training that prevents generator from producing unreasonable summaries of source text. We found that if we teacher-forced generator too frequently, generator would overfit on training data since we only use very few labeled data on semi-supervised training.

The performance of semi-supervised model in English Gigaword regarding available labeled data is shown in Fig. 6. The horizontal axis is the number of labeled documents used in the experiments, while the vertical axis for Fig. 6 (a) and (b) are ROUGE-1 and ROUGE-2 respectively. The green curve is the results of supervised learning, and the red and blue curves are semi-supervised learning with different approaches. With the same amounts of labeled data, the performances of semi-supervised training are always better than supervised training. With only 100K labeled data, the ROUGE score of semi-supervised training using adversarial REINFORCE even slightly outperformed supervised training with whole labeled data. This shows that with the proposed approach, we need only 2.6% of labeled data to achieve the same performance as before (100K v.s. 3.8M). The complete results for semi-supervised learning in both datasets are shown in Appendix B.

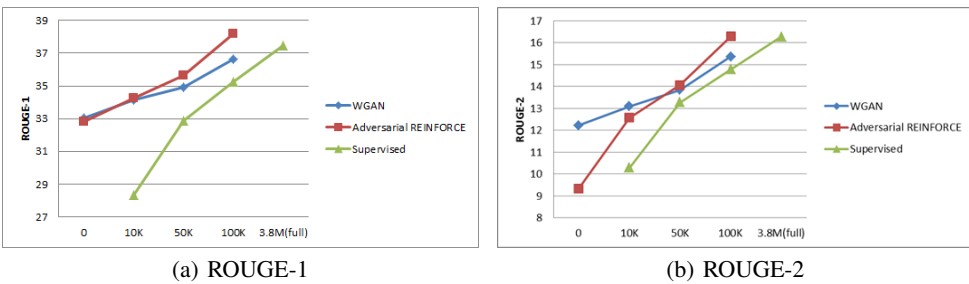

(a) ROUGE-1                         (b) ROUGE-2

Figure 6: Semi-supervised results in English Gigaword. With the same amount of labeled data, the performances of semi-supervised training are always better than supervised training.

## 7.5 GAN TRAINING

| Corpus | Method | ROUGE-1 | ROUGE-2 | ROUGE-L |
|--------|--------|---------|---------|---------|
| Chinese | with self-critic | 40.053 | 26.126 | 37.118 |
| | without self-critic | 37.549 | 24.181 | 35.160 |
| English | with self-critic | 32.826 | 9.332 | 28.727 |
| | without self-critic | 31.104 | 9.249 | 28.592 |

Table 3: Adversarial REINFORCE with/without self-critic with unsupervised training.

The two GAN training methods are not comparable as their settings are quite different, but we can still discuss the advantages and disadvantages of these two methods. When training with feeding output layer to discriminator, convergence is faster. This method sharpens the distribution at an early stage in training because it directly evaluates the distance between the generator's continuous distribution and the real data's discrete distribution data. However, this cause generator to converge to a not very good place.

In fact, adversarial REINFORCE is sensitive to initialization parameters. Adversarial REINFORCE requires better initialization for exploration; otherwise, with so many actions whose number is equal to vocabulary size to choose, it is extremely difficult to train generator from scratch. In semi-supervised training, since we pre-trained generator with labeled data, the generator was better initialized, therefore adversarial REINFORCE performed better. To support this idea, in Fig. 6, we compare the performance of two methods regarding labeled data. The result implies that with more labeled data, our proposed adversarial REINFORCE method performs better. In order to evaluate the performance of proposed self-critic baseline trick mentioned in Section 5.2.2, we compared the performance of our model with and without this baseline trick in Table 3. For the experiments without the self-critic baseline trick, we replaced $s_i - s_{i-1}$ in Section 5.2.2 with $s_i$. We found that the performance degraded without self-critic.

## 8 CONCLUSION AND FUTURE WORK

Using GAN, we propose a model that encodes text as a human-readable summary, learned without document-summary pairs. Promising results are obtained on both Chinese and English corpora. In future work, we hope to explore more techniques for natural language generation using GAN. Moreover, we hope to use extra discriminators to control the style and sentiment of the generated summaries.

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

## A   MODEL PRE-TRAINING

As we found that the different pre-training methods for the generator influenced final performance dramatically in all of the experiments, we felt it was important to find a proper unsupervised pre-training method to help the machine grasp semantic meaning. We used the different pre-training strategies described below.

- Chinese Gigaword: Given the previous $i - 1$ sentences $sent_0, sent_1, ..., sent_{i-1}$ from the source text, the generator predicted the next sentence $sent_i$ in the source text as its pre-training target. If more than 50% of the words in sentence $sent_i$ did not appear in the given text, we filtered out this pre-training sample pair. This pre-training method allowed the generator to capture the important semantic meanings of the source text.

- English Gigaword: As the length of the source texts in English Gigaword dataset is comparatively short, it is difficult to split the last sentence from the source text; hence the previous pre-training method on Chinese Gigaword is not appropriate for this dataset. To properly initialize the set, we randomly selected 6 to 11 consecutive words in the source text, after which we randomly swapped 70% of the words in the source text. Given text with incorrect word arrangements, the generator predicted the selected words in the correct arrangement. We pre-trained in this way because we expect the generator to initialize with a rough language model. In Chinese Gigaword we also conducted experiments on pre-training in this manner, but the results were not as good as those shown in the part (C) of Table 1. We also used the retrieved paired data in row (B-1) in Table 2 to pre-train generator. However, pre-training generator with this method doesn't yield results better than those in Table 2.

# B  SEMI-SUPERVISED LEARNING

Semi-supervised learning experiments were conducted with 10K, 50K, 100K labeled data in both datasets. We conducted teacher-forcing on generator every 30, 12, 7 unsupervised updates with 10K, 50K, 100K labeled data respectively. The complete results for semi-supervised learning are shown in Tables 4 and 5.

| | | ROUGE-1 | ROUGE-2 | ROUGE-L |
|---|---|---|---|---|
| Unsupervised | WGAN | 37.803 | 24.460 | 35.116 |
| | Adversarial REINFORCE | 40.053 | 26.126 | 37.118 |
| Semi-supervised (10K labeled) | WGAN | 42.359 | 27.192 | 38.467 |
| | Adversarial REINFORCE | 43.109 | 28.626 | 40.202 |
| Semi-supervised (50K labeled) | WGAN | 43.989 | 29.012 | 40.764 |
| | Adversarial REINFORCE | 44.706 | 29.872 | 41.737 |
| Semi-supervised (100K labeled) | WGAN | 45.642 | 31.475 | 42.711 |
| | Adversarial REINFORCE | 46.216 | 31.980 | 43.522 |
| Supervised | | 48.664 | 33.907 | 45.685 |

Table 4: Semi-supervised learning in Chinese Gigaword with different amounts of labeled data (10K, 50K, 100K).

| | | ROUGE-1 | ROUGE-2 | ROUGE-L |
|---|---|---|---|---|
| Unsupervised | WGAN | 33.043 | 12.222 | 30.121 |
| | Adversarial REINFORCE | 32.826 | 9.332 | 28.727 |
| Semi-supervised (10K labeled) | WGAN | 34.127 | 13.087 | 31.451 |
| | Adversarial REINFORCE | 34.239 | 12.570 | 31.834 |
| Semi-supervised (50K labeled) | WGAN | 34.937 | 13.838 | 32.372 |
| | Adversarial REINFORCE | 35.642 | 14.057 | 32.983 |
| Semi-supervised (100K labeled) | WGAN | 36.615 | 15.363 | 33.682 |
| | Adversarial REINFORCE | 38.213 | 16.279 | 35.137 |
| Supervised | | 37.469 | 16.272 | 35.175 |

Table 5: Semi-supervised learning in English Gigaword with different amounts of labeled data (10K, 50K, 100K).

## C  EXAMPLES

From Fig. 7 to 12, we show more examples.

| Source Text: |
| --- |
| former zambian president kenneth kaunda appeared in court monday on charges of holding an illegal rally , declaring that he would continue to fight the `` oppressive regime '' of president fredrick chiluba . |

| Ground Truth: | (A)Supervised Result: |
| --- | --- |
| former zambian president in court for illegal assembly | kaunda to continue to fight chiluba |
| (C-1)Pretrained Generator: | (D-1)Pretrained Generator: |
| president kenneth chiluba appeared in court monday on charges of | kaunda declaring that he would continue to fight the regime says |
| (C-2)WGAN: | (C-3)Adversarial REINFORCE: |
| zambian kenneth kaunda in court charges of holding illegal rally | former zambian president to continue illegal rally fight |
| (E-1)WGAN : | (E-2)Adversarial REINFORCE: |
| former zambian president kenneth kaunda appeared in court of illegal rally | he appeared in court he would continue to fight the regime |

Figure 7: Real example from our model in English Gigaword. In part (E-2), due to transfer learning, the summary sentence begins with word *he*, which never appears in the English Gigaword summary sentences.

| Source Text: |
| --- |
| hong kong tourist association -lrb- hkta -rrb- said wednesday it regretted having placed an advertisement thanking sponsors of last week 's lunar new year parade which killed one man and left ## others injured . |

| Ground Truth: | (A)Supervised Result: |
| --- | --- |
| hong kong tourist body expresses regret over advertisement | hong kong tourist regrets having placed advertisement sponsors |
| (C-1)Pretrained Generator: | (D-1)Pretrained Generator: |
| hong kong said wednesday it regretted an advertisement | one man was killed in the head of the sponsors of the said |
| (C-2)WGAN: | (C-3)Adversarial REINFORCE: |
| hong kong tourist association regretted having placed advertisement sponsors | hong kong tourist killed advertisement sponsors of last year 's lunar parade |
| (E-1)WGAN : | (E-2)Adversarial REINFORCE: |
| hong kong tourist association regretted having placed an advertisement sponsors | it regretted having placed advertisement sponsors of last week 's lunar new year |

Figure 8: Real example from our model in English Gigaword. The sentence grammar in part (C-3) is correct, but the semantics are incorrect.

| Source Text: experts from iran and the un nuclear watchdog met thursday in vienna to discuss tehran 's plans to resume atomic fuel research , an official from the agency said . | |
|---|---|
| **Ground Truth:** iranian experts meet with un nuclear watchdog | **(A)Supervised Result:** iran un nuclear watchdog discuss tehran |
| **(C-1)Pretrained Generator:** to discuss tehran 's nuclear research watchdog | **(D-1)Pretrained Generator:** . official says experts from iran 's president 's watchdog met |
| **(C-2)WGAN:** experts iran to un nuclear watchdog in vienna | **(C-3)Adversarial REINFORCE:** experts discuss un nuclear watchdog plans to resume tehran |
| **(E-1)WGAN :** experts from iran and the un nuclear watchdog met | **(E-2)Adversarial REINFORCE:** experts from iran and the un nuclear watchdog met |

Figure 9: Real example from our model in English Gigaword.

| Source Text: european shares fell monday , pressured by higher crude prices after oil giant bp said it will shut down a key production field and on caution before a u.s. interest-rate decision . | |
|---|---|
| **Ground Truth:** european stocks end lower | **(A)Supervised Result:** european stocks fall on bp decision |
| **(C-1)Pretrained Generator:** prices fell monday after caution on caution | **(D-1)Pretrained Generator:** oil giant bp says higher crude prices on caution |
| **(C-2)WGAN:** european shares shut down higher after oil | **(C-3)Adversarial REINFORCE:** european shares shut down after higher oil production |
| **(E-1)WGAN :** european shares fell by higher crude prices after oil | **(E-2)Adversarial REINFORCE:** by higher crude prices after oil giant and on caution before it will shut |

Figure 10: Real example from our model in English Gigaword.

| Source Text: | |
|---|---|
| russian foreign minister igor ivanov urged iran to be open about its nuclear programs during a meeting with his iranian counterpart at the united nations , the foreign ministry said tuesday . | |
| **Ground Truth:** russian minister urges iran to be open about nuclear programs | **(A)Supervised Result:** russian fm urges iran to be open about nuclear programs |
| **(C-1)Pretrained Generator:** ivanov urged iran to be open its nuclear programs during | **(D-1)Pretrained Generator:** russian foreign minister igor ivanov says meeting will be open |
| **(C-2)WGAN:** russian igor to be open about nuclear programs during meeting | **(C-3)Adversarial REINFORCE:** russian foreign minister to open iran meeting about its nuclear programs |
| **(E-1)WGAN :** russian foreign minister igor ivanov urged iran to be open about nuclear programs | **(E-2)Adversarial REINFORCE:** new foreign feature urges iran to be open about its nuclear programs |

Figure 11: Real example from our model in English Gigaword.

| Source Text: | |
|---|---|
| the bewildering fight between the government and telemarketers over the national do-not-call list took another turn when a second federal agency said it would enforce the program , promising that consumers would soon see some reduction in telephone sales pitches . | |
| **Ground Truth:** fcc steps in to enforce do-not-call list ; bush signs new law to support program | **(A)Supervised Result:** fight against consumers |
| **(C-1)Pretrained Generator:** the second list in telephone sales that would enforce | **(D-1)Pretrained Generator:** , promising that consumers would enforce turn |
| **(C-2)WGAN:** fight between government over national list took turn | **(C-3)Adversarial REINFORCE:** fight between government pitches see another reduction |
| **(E-1)WGAN :** the fight between the government and over the national list took another turn | **(E-2)Adversarial REINFORCE:** fight between government and over national list took another turn when a second federal |

Figure 12: Real example from our model in English Gigaword.

