# OpenReview forum: "Learning to Encode Text as Human-Readable Summaries using Generative Adversarial Networks"
_ICLR.cc/2018/Conference — Reject_

### Official Review · AnonReviewer2 · 2017-11-28
**GANs for text, with text latent variables**

**Rating:** 5
**Confidence:** 4

**Review:**

This paper proposes a model for generating long text strings given shorter text strings, and for inferring suitable short text strings given longer strings. Intuitively, the inference step acts as a sort of abstractive summarization. The general gist of this paper is to take the idea from "Language as a Latent Variable" by Miao et al., and then change it from a VAE to an adversarial autoencoder. The authors should cite "Adversarial Autoencoders" by Makzhani et al. (ICLR 2016).

The experiment details are a bit murky, and seem to involve many ad-hoc decisions regarding preprocessing and dataset management. The vocabulary is surprisingly small. The reconstruction cost is not precisely explained, though I assume it's a teacher-forced conditional log-likelihood (conditioned on the "summary" sequence). The description of baselines for REINFORCE is a bit strange -- e.g., annealing a constant in the baseline may affect variance of the gradient estimator, but the estimator is still unbiased and shouldn't significantly impact exploration. Similar issues are present in the "Self-critical..." paper by Rennie et al. though, so this point isn't a big deal.

The results look decent, but I would be more impressed if the authors could show some benefit relative to the supervised model, e.g. in a reasonable semisupervised setting. Overall, the paper covers an interesting topic but could use extra editing to clarify details of the model and training procedure, and could use some redesign of the experiments to minimize the number of arbitrary (or arbitrary-seeming) decisions.

---

> ### Author Response · Authors · 2017-12-31
> **Response to AnonReviewer2**
>
> We really appreciate your comment and suggestions.  The paper has been carefully revised. All the page numbers below refer to the revised version.
>
> 1. We have cited “Adversarial Autoencoder” by Makhzani et al.
>
> 2. Data Preprocessing:
> In Chinese Gigaword corpus, the arbitrary decisions regarding on data preprocessing aim to filter out some bad training examples. We conducted all experiments including baseline experiment on same set of pre-processed data. Hence, we still can compare our model to baseline models. In English Gigaword corpus, we simply use the training set pre-processed by previous work (A Neural Attention Model for Abstractive Sentence Summarization by Rush et al. 2015) and don’t do any further preprocess.
>
> 3. Vocabulary size:
> The reason that the vocabulary size  in Chinese corpus is extremely small (4K) is that the text unit we use is Chinese character instead of  Chinese word. In English corpus, the vocabulary size we used is 15K which is in a reasonable range.
>
> 4.  Reconstruction cost:
> The reconstruction cost is conditioned on generated summary sequence and is teacher forced by source text. We add more description to clarify this. Please refer to the first 4 line in P4.
>
> 5. Semi-supervised training:
> In semi-supervised training, we first pre-trained the generator with few labeled data. Then, we conducted teacher forcing with labeled data every several unsupervised training steps. We evaluate the performance of our model with regard to the number of labeled data. It’s worth mentioning that In English Gigaword corpus, with only 100K labeled data, semi-supervised training even slightly outperforms supervised-training with full labeled data. Please refer to the results in Figure 6 (P10). Furthermore, we also discussed the performance of our proposed adversarial REINFORCE in Section 7.5(P10) with regard to number of labeled data in semi-supervised learning.
>
> 6. Issues of ad-hoc decisions:
> We found that in the semi-supervised scenario if we pretrain generator with few labeled data, ad-hoc decisions regarding on pre-training generator are not necessary. However, in completely unsupervised setting, we still not come up with a proper method to prevent ad-hoc decisions on pre-training generator.
>
> 7. Clarification of details of the model and training procedure:
> We have made some extra editing to clarify the details of model and training. The details are provided in Section 6(P6).

---

> > ### Comment · AnonReviewer2 · 2018-01-12
> > **paper is improved**
> >
> > Thank you for revising the paper. It is easier to read now, though later sections still seem less edited than the beginning.
> >
> > For the semisupervised experiments a more appropriate baseline would be a likelihood-based equivalent of your technique, e.g. the "dual" training by He et al. 2016 in "Dual Learning for Machine Translation".

---

> > > ### Author Response · Authors · 2018-01-17
> > > **Thank you for your review**
> > >
> > > Thank you for reading the paper again and giving us comment. We will improve the writing of later sections. If we want to apply dual learning in this text summarization task, the training is not only on “source text -> summary -> source text”, but also on “summary -> source text -> summary”. In the “source text -> summary -> source text” path, reconstructor (summary -> source text) produces source text with teacher forcing because the source text is known. However, in the “summary -> source text -> summary” path, it’s difficult for reconstructor to produce source text from summaries without teacher-forcing (due to the unsupervised update) since source text is long. Hence, we do not consider this baseline in the first place. But if possible to modify the paper in the future, we will compare duel learning with our results on semi-supervised training.

---

### Official Review · AnonReviewer1 · 2017-11-28
**LEARN TO ENCODE TEXT AS COMPREHENSIBLE SUMMARY BY GENERATIVE ADVERSARIAL NETWORK**

**Rating:** 4
**Confidence:** 4

**Review:**

Summary: In this work, the authors propose a text reconstructing auto encoder which takes a sentence as the input sequence and an integrated text generator generates another version of the input text while a reconstructor determines how well this generated text reconstructs the original input sequence. The input to the discriminator (as real data) is a sentence that summarizes the ground truth sentences (rather than the ground truth sentences themselves). The experiments are conducted in two datasets of English and Chinese corpora.

Strengths:
The proposed idea of generating text using summary sentences is new.
The model overview in Figure 1 is informative.
The experiments are conducted on English and Chinese corpora, comparison with competitive baselines are provided.

Weaknesses:
The paper is poorly written which makes it difficult to understand. The second paragraph in the introduction is quite cryptic. Even after reading the entire paper a couple of times, it is not clear how the summary text is obtained, e.g. do the authors ask annotators to read sentences and summarize them? If so, based on which criteria do the annotators summarize text, how many annotators are there? Similarly, if so this would mean that the authors use additional supervision than the compared models. Please clarify how the summary text is obtained.

In footnote 1, the authors mention “seq2seq2seq2” term which they do not explain anywhere in the text.

No experiments that generate raw text (without using summaries) are provided. It would be interesting to see if GAN learns to memorize the ground truth sentences or generates sentences with enough variation.

In the English Gigaword dataset the results consistently drop compared to WGAN. This behavior is observed for both the unsupervised setting and two versions of transfer learning settings. There are too few qualitative results: One positive qualitative result is provided in Figure 3 and one negative qualitative result is provided in Figure 4. Therefore, it is not easy for the reader to judge the behavior of the model well.

The choice of the evaluation metric is not well motivated. The standard measures in the literature also include METEOR, CIDER and SPICE. It would be interesting to see how the proposed model performs in these additional criteria. Moreover, the results are not sufficiently discussed.

As a general remark, although the idea presented in this paper is interesting, both in terms of writing and evaluation, this paper has not yet reached the maturity expected from an ICLR paper. Regarding writing, the definite and indefinite articles are sometimes missing and sometimes overused, similarly most of the times there is a singular/plural mismatch. This makes the paper very difficult to read. Often the reader needs to guess what is actually meant. Regarding the experiments, presenting results with multiple evaluation criteria and showing more qualitative results would improve the exposition.

Minor comments:
Page 5: real or false —> real or fake (true or false)
	     the lower loss it get —> ?

---

> ### Author Response · Authors · 2017-12-31
> **Response to AnonReviewer1**
>
> Thank you for your thorough read of the paper and pointing out our defects. The paper has been carefully revised. All the page numbers below refer to the revised version.
>
> Major revisions:
> 1. Writing:
> We acknowledge that there are some defects in original version of paper which make readers difficult to understand. We have revised the grammatical errors in the paper. Because all the authors are not English naïve speakers, we hired an English native speaker with Computer Science PhD. to help us polish the English writing.
>
> 2. How to obtain summaries:
> The documents used in the study are from news. The titles of the documents are considered as the summaries. This is a typical setup in the study of summarization. We add footnote 1 for the above description (P2).
>
> 3. Clarification of the core idea:
> We are very sorry that the original version of introduction is misleading. The purpose of the work is to generate summaries from an article, not to exploit summaries to better generate articles. First, instead of encoding a sentence into another version of sentence, our text auto-encoder encodes long text into short text while reconstructor tries to reconstruct long text from encoded short text. The discriminator regularizes the latent representations encoded by encoder (generator in our paper) to be human-readable summaries. The short text encoded by generator can be considered as summary of long text, and thus unsupervised text summarization is achieved. The discriminator can use any human-written sentences as real data. Hence, there is no need of human annotators.
> In the real implementation, instead of using general human-written sentences, we use the sentences from the titles of the documents as real data for discriminator for better performance. However, the titles do not have to be paired with the training documents (for example, in Section 7.3(P8), we can use documents from Gigaword and titles from CNN/Diary), so the training in unsupervised.
> We have re-written the introduction, especially the second paragraph. We also add an overview figure to clearly describe the basic idea. Please refer to Fig. 1 (P2).
>
> For specific points:
> 1. “seq2seq2seq” in footnote:
> In the typical seq2seq model, the input sequence is compressed into a vector and then back to another sequence. In our model, the input long sequence is first compressed into shorter sequence, and the model uses the short sequence to generate the long sequence. Hence, we called it “seq2seq2seq” model. This footnote is removed from the revised version.
>
> 2. Experiments about text generation:
> The target of this work is to generate short text as summary of input document instead of generating raw text. The generator never sees the summaries of the documents, so it cannot memorize the summaries.
>
> 3. Comparison to original WGAN:
> As mentioned in your review, in English Gigaword, compared to WGAN, the performance of our proposed adversarial REINOFRCE consistently drops both in unsupervised learning and transfer learning. However, after conducting semi-supervised training experiments, we found that with more labeled data available, adversarial REINFORCE is better than WGAN. We compared the performance of two models regarding available labeled data. Please find the results in Fig. 6 (P9). The full discussion of the two models is in Section 7.5 (P10). We also found that self-critic in Section 5.2.2 is helpful. The results are shown in Table 3 (P10).
>
> 4. More examples:
> To make reader better judge the proposed model, besides Fig. 3 and 4, we have more results in the appendix. Please refer to Fig. 7 to 12 (P15 - 17).
>
> 5. Evaluation metric:
> We know that ROUGE is not a perfect evaluation for summarization, but ROUGE is widely used to evaluate the generated summaries. In the previous work (A Neural Attention Model for Abstractive Sentence Summarization by Rush et al. 2015; Abstractive text summarization using sequence-to-sequence rnns and beyond, Nallapati et al. 2016; Abstractive Sentence Summarization with Attentive Recurrent Neural Networks, Chopra et al 2016), ROUGE is the only major evaluation measure used to evaluate the quality of the summaries.

---

### Official Review · AnonReviewer3 · 2017-11-28
**Simple but useful extension of recent works, missing important experiments**

**Rating:** 6
**Confidence:** 4

**Review:**

TL;DR of paper: Generating summaries by using summaries as an intermediate representation for autoencoding the document. An encoder reads in the document to condition the generator which outputs a summary. The summary is then used to condition the decoder which is trained to output the original document. An additional GAN loss is used on the generator output to encourage the output to look like summaries -- this procedure only requires unpaired summaries. The results are that this procedure improves upon the trivial baseline  but still significantly underperforms supervised training.

This paper builds upon two recent trends:  a) cycle consistency, where f(g(x)) = x, which only requires unpaired data (i.e., CycleGAN), and (b) encoder-decoder models with a sequential latent representation (i.e., "Language as a latent variable" by Miao and Blunsom). A similar idea has also been explored by He et al. 2016 in "Dual Learning for Machine Translation". Both CycleGAN and He et al. 2016 are not cited. The key difference between this paper and He et al. 2016 is the use of GANs so only unpaired summaries are needed.

The idea is a simple but useful extension of these previous works. The problem set-up of unpaired summarization is not particularly compelling, since summaries are typically found paired with their original documents. It would be more interesting to see how well it can be used for other textual domains such as translation, where a lot of unpaired data exists (some other submissions to ICLR tackle this problem). Unsurprisingly, the proposed method requires a lot of twiddling to make it work since GANs, REINFORCE, and pretraining are necessary.

A key baseline that is missing is pretraining the generator as a language model over summaries. The pretraining baseline in the paper is over predicting the next sentence / reordering, but this is an unfair comparison since the next sentence baseline never sees summaries over the course of training. Without this baseline, it is hard to tell whether GAN training is even useful. Another experiment missing is seeing whether joint supervised-GAN-reconstruction training can outperform purely supervised training. What is the performance of the joint training as the size of the supervised dataset is varied?

This paper has numerous grammatical and spelling errors throughout the paper (worse, the same errors are copy-pasted everywhere). Please spend more time editing the paper.

---

> ### Author Response · Authors · 2017-12-31
> **Response to AnonReviewer3**
>
> Thank you for giving us some helpful suggestions. To reply your comment, the paper has been carefully revised. All the page numbers below refer to the revised version.
>
> We made the following modifications of paper:
> 1. We have cited the papers of Cycle GAN and He et al. 2016 mentioned in your comment.
>
> 2.  Problem setup:
> It is true that in news domain, it is relatively easy to find document-summary pairs because usually people consider the news titles as summaries. However, for the domains like lecture recording, we think collecting labelled data is not trivial. Therefore, it is worth to study unsupervised abstractive summarization. In this paper, we still conduct the experiments on the news domain because the ground truth is available for evaluation. In the future, we can extend to other domains in which collecting label data is changeling.
>
> 3. Pre-training generator as language model over summaries:
> The model architecture of generator is a hybrid pointer network in which decoder selects part of the words from the generator input text. Hence, it’s difficult to train the generator as language model of summary without input text. We came up with another method that solves this problem. Given a set of unpaired documents and summaries, we used an unsupervised approach to match each document with its most relevant  summaries. We represented each document and each summary as tf-idf (term frequency–inverse document frequency) vectors. Each document is matched to the summary whose vector has the largest cosine similarity with the document vector.
> We further used the retrieved paired data to train generator and regarded its performance as baseline. With this method, generator can be roughly initialized with a language model of summaries. The ROUGE scores obtained in this approach is shown in row (B-2) of Table 2 (P8) Then we further improve the generator pre-trained in this way by the proposed unsupervised approach. However, with generator pre-trained by this method, we do not obtain the results better than the ones in Table 2.
>
> 4. Semi-supervised training:
> In semi-supervised training, we first pre-trained the generator with few labeled data. Then, we conducted teacher forcing with labeled data every several unsupervised training steps. We evaluated the performance of our model with regard to the number of labeled data. It’s worth mentioning that In English Gigaword corpus, with only 100K labeled data, semi-supervised training even slightly outperforms supervised-training with full labeled data. Please refer to the results in Figure 6 (P10) and Appendix B(P14). Furthermore, we also discussed the performance of our proposed adversarial REINFORCE in Section 7.5(P10) with regard to number of labeled data in semi-supervised learning.
>
> 5. Writing:
> Because all the authors are not English native speaker, we hired an English native speaker with Computer Science PhD. to help us polish the English writing.

---

### Decision · Program_Chairs · 2018-01-29
**ICLR 2018 Conference Acceptance Decision**

**Decision:**

Reject

**Comment:**

As expressed by most reviewers, the idea of the paper is interesting:  using summarization as an intermediate representation for an auto encoder.  In addition, a GAN is used on the generator output to encourage the output to look like summaries.  They just need unpaired summaries.  Even if the idea is interesting, from the committee's perspective, important baselines are missing in the experimental section:  why would one choose to use this method if it is not competitive with other baselines that have proposed work in this vein?  One reviewer brings up the point that the method is significantly worse than a supervised baseline.  Moreover, the authors mention the work of Miao and Blunsom, but could have used one of their experimental setups to show that at least in the semi-supervised scenario, this work empirically performs as well or better than that baseline.